# Siponimod Modulates the Reaction of Microglial Cells to Pro-Inflammatory Stimulation

**DOI:** 10.3390/ijms232113278

**Published:** 2022-10-31

**Authors:** Joel Gruchot, Ferdinand Lein, Isabel Lewen, Laura Reiche, Vivien Weyers, Patrick Petzsch, Peter Göttle, Karl Köhrer, Hans-Peter Hartung, Patrick Küry, David Kremer

**Affiliations:** 1Department of Neurology, Medical Faculty, University Hospital Düsseldorf, Heinrich Heine University, Moorenstraße 5, D-40225 Dusseldorf, Germany; 2Biological and Medical Research Center (BMFZ), Medical Faculty, Heinrich-Heine-University, D-40225 Dusseldorf, Germany; 3Brain and Mind Center, University of Sydney, Sydney, NSW 2050, Australia; 4Department of Neurology, Palacky University Olomouc, 77146 Olomouc, Czech Republic

**Keywords:** sphingosine 1-phosphate receptor signalling, multiple sclerosis, neurodegeneration, modulation, polarization

## Abstract

Siponimod (Mayzent^®^), a sphingosine 1-phosphate receptor (S1PR) modulator which prevents lymphocyte egress from lymphoid tissues, is approved for the treatment of relapsing-remitting and active secondary progressive multiple sclerosis. It can cross the blood–brain barrier (BBB) and selectively binds to S1PR1 and S1PR5 expressed by several cell populations of the central nervous system (CNS) including microglia. In multiple sclerosis, microglia are a key CNS cell population moving back and forth in a continuum of beneficial and deleterious states. On the one hand, they can contribute to neurorepair by clearing myelin debris, which is a prerequisite for remyelination and neuroprotection. On the other hand, they also participate in autoimmune inflammation and axonal degeneration by producing pro-inflammatory cytokines and molecules. In this study, we demonstrate that siponimod can modulate the microglial reaction to lipopolysaccharide-induced pro-inflammatory activation.

## 1. Introduction

Myelin sheaths in the human central nervous system (CNS) stabilize, trophically support and electrically insulate axons but are destroyed in demyelinating diseases such as multiple sclerosis (MS). As a result, saltatory signal conduction is interrupted and axonal damage occurs manifesting itself in various clinical symptoms [1]. Siponimod (BAF312), an orally administered sphingosine-1-phophate receptor (S1PR) modulator binding selectively to S1PR1 and S1PR5, reduces relapse rate and inflammatory disease activity in relapsing remitting MS (RRMS) [2]. In addition, it was shown to slow down disease progression and brain atrophy in secondary progressive MS (SPMS) [3] pointing to potential anti-neurodegenerative properties. Furthermore, several studies have demonstrated that siponimod reduces myelin loss in organotypic slice cultures [4] and prevents synaptic loss in the MS animal model myelin oligodendrocyte glycoprotein-induced experimental autoimmune encephalitis (MOG-EAE) [5]. While the exact underlying mechanisms are still unclear, microglia (MG) might be highly relevant in this context [6]. MG are glial cells in the CNS, play a key role in MS-associated inflammatory processes [7] and express both S1PR1 and S1PR5 [8]. While they can adopt an anti-inflammatory phenotype in which they clear myelin debris and promote regeneration, i.e., exert neuroprotective effects [8,9], they are also able to produce pro-inflammatory cytokines in response to MS-related CNS injury [10,11]. Thereby the inflammatory process is upheld by cytokine-mediated activation of other immune cells ultimately leading to myelin destruction and subsequent neurodegeneration. Of note, MG can switch back and forth between pro- and anti-inflammatory phenotypes based on the underlying disease pathology—a process called polarization. Taken together, it is therefore conceivable that via a modulation of MG polarization siponimod could potentially exert effects on ongoing neurodegenerative processes in MS.

## 2. Results

### 2.1. Siponimod Modulates Microglial Morphology and Actin Filament Organization

In order to better understand the effects of siponimod on brain-resident cells in the context of MS, we investigated whether siponimod can modulate the behaviour of primary rat microglia in an inflammatory milieu. To this end, we used lipopolysaccharide (LPS) as a pro-inflammatory cue. We stimulated cultured primary rat microglia with either 10/50 µM siponimod or 100 ng/mL LPS and compared their reaction to a simultaneous stimulation with both reagents. As respective controls, we used DMSO for siponimod and ddH_2_O for LPS, either alone or in combination (Figure 1). With regard to the seemingly high siponimod concentrations used in our experiments performed in 10% foetal calf serum-containing medium, several studies have demonstrated > 99.9% protein binding for siponimod [12]. Moreover, other studies have shown that the 10 and 50 µM (total) drug concentrations used by us are indeed observed in the CNS of EAE-mice receiving efficacious treatment with siponimod [13]. In greater detail, Bigaud and colleagues showed that feeding mice with a diet containing 100 mg per kg pellet results in total brain siponimod levels exceeding 10 µM. Using Iba1 staining for visualization we found that LPS stimulation increases the number of microglia (Figure 1A,B blue bars) compared to control stimulated cells (Figure 1B white bars). At the higher concentration of 50 µM but not at 10 µM, siponimod treatment (Figure 1B green bars) also resulted in an increase in cell number compared to control cells (Figure 1B white bars). However, co-stimulation with LPS (Figure 1B red bars) resulted in a significant cell number reduction compared to LPS-stimulated cells (Figure 1B blue bars).

It has been well-described previously that activated microglia change their morphology and cytoskeletal actin organization [14]. Therefore, we quantified the size of microglial cells, confirming that LPS induces microglia to significantly increase their mean cell area (Figure 1A,C blue bars) in comparison to control cells (Figure 1C white bars). At the higher concentration of 50 µM, siponimod alone slightly altered cell morphology in comparison to control cells (Figure 1C green bars). Most importantly, however, we observed that co-stimulation of microglia with siponimod at a concentration of 50 µM and LPS significantly reduced the LPS-induced increase in mean cell area (Figure 1C red bar).

In order to evaluate actin cytoskeleton organization, we used Alexa488 coupled phalloidin to specifically stain filamentous and non-filamentous forms of actin (Figure 1D). As expected, control cells displayed a homogenous distribution of mostly non-filamentous forms of actin. In contrast, LPS-stimulated microglia featured strong filamentous forms of actin. However, microglia stimulated with 50 µM siponimod alone no longer showed any filamentous actin at all and when co-stimulated with LPS the actin organization was again homogenously distributed and only very few cells showed filamentous actin.

### 2.2. Siponimod Modulates iNOS Protein Expression

Nitric oxide (NO) is a free radical found at high concentrations in inflammatory multiple sclerosis (MS) lesions. This is based on an increased expression of inducible form of nitric oxide synthase (iNOS) in cells such as microglia, myeloid cells and astrocytes. NO plays a role in the disruption of the blood–brain barrier (BBB), oligodendrocyte injury, demyelination, axonal degeneration, mitochondrial dysfunction and impairment of axonal conduction [15,16]. We therefore sought to investigate whether siponimod may also modulate iNOS expression (Figure 2). Using the same stimulation scheme as delineated above, we found that LPS alone lead to a significant increase in iNOS protein expression (Figure 2A blue columns, Figure 2B) in comparison to controls (Figure 2A white bars, Figure 2B). However, at both concentrations of 10 and 50 µM siponimod significantly reduced LPS-induced microglial iNOS expression at protein level (Figure 2A red columns, Figure 2B).

### 2.3. Siponimod Modulates Microglial Cytokine Gene Expression

Using the same stimulation scheme, we next investigated if siponimod also modulates microglial cyto- and chemokine gene expression patterns relevant in the course of MS (Figure 3). We found that in parallel to our morphology experiments (see Figure 1), LPS induced microglia to significantly increase the expression of the pro-inflammatory factors, tumor necrosis factor-α (Tnf/TNFα; Figure 3A blue bar) and interleukin-1β (Il1b/IL-1β; Figure 3E blue bar) in comparison to control cells (Figure 3A,E white bars). Siponimod at a concentration of 50 µM alone did not alter microglial gene expression in comparison to control cells (Figure 3A,E green bars). However, siponimod significantly reduced the LPS-induced increase in TNFα and IL-1β (Figure 3A,E red bars) which we corroborated at protein levels (Figure 3B,F red bars). Increased TNFα production can be found in active MS lesions, sera and the cerebrospinal fluid (CSF) of MS patients. It contributes to neuronal death as well as axonal damage and also correlates with the severity and progression of the disease [8,17,18,19]. IL-1β is a pro-inflammatory cytokine which induces excitotoxic neurodegeneration. In addition, there is an association between IL-1β CSF levels and disability progression in RRMS patients [20]. We also found that siponimod prevented an LPS-induced induction of interferon-β (Ifnb/IFNβ, Figure 2C red bar) and of interleukin-10 (Il10/IL-10, Figure 2D red bar) in comparison to control-stimulated cells. In addition, the secretion of IL-10 and IFNβ proteins was significantly decreased upon siponimod stimulation (Figure 3D,H red bars) compared to LPS alone (Figure 3D,H blue bars). IL-10 is a potent anti-inflammatory cytokine that is able to suppress the synthesis of pro-inflammatory cytokines such as IFNγ [21,22,23]. IFNβ possesses both pro- and anti-inflammatory properties and was the first substance to be used clinically as a drug against RRMS [24]. Of note, non-polarized microglial cells exhibited either very low or below-detection-limit levels of corresponding secreted cytokines.

### 2.4. Siponimod Modulates Immunological Signature in Pro-Inflammatory Triggered Microglial Cells

Since we found that siponimod exerted an anti-inflammatory effect on LPS-stimulated microglial cells, bulk RNA sequencing was performed to further describe the resulting microglial gene expression signature. Using the same scheme as described above, we compared the transcriptomes of cells stimulated with the combination of LPS and siponimod with cells stimulated with LPS alone. In order to assess significantly dysregulated genes, plotting the log_2_-fold-change (FC) against the log_10_-adjusted *p*-value (false discovery rate, FDR, Figure 4A) was performed. This resulted in the identification of Clec5a, Fst, Cd9, Clec4d, Cdkn1c and Cd34 as the genes most significantly upregulated in the presence of LPS and siponimod compared to LPS stimulation alone. Furthermore, Cxcl11, Angptl4, Cd38 and Fscn1 were identified as most significantly downregulated. Using a fold-change difference of ±1.5 and adjusted *p*-value ≤ 0.05 to determine differentially expressed genes (DEGs) in the siponimod and LPS-treated group compared to the LPS alone group, we identified 3425 DEGs (Figure 4B), of which 1658 were significantly upregulated and 1767 were significantly downregulated. To further characterize this microglial signature, we performed gene enrichment analysis (GO Biol. Processes, Figure 4C,D), identifying “response to unfolded protein”, “response to endoplasmatic reticulum stress” and “response to starvation” as the most significant pathways related to upregulated DEGs. On the other hand, downregulated DEGs were enriched in “positive regulation of immune response”, “leukocyte activation”, “regulation of cytokine production” and “innate immune response”. This demonstrated that the immunological pathways activated by LPS in microglia were downregulated in response to siponimod. The top 20 downregulated genes of each of these four biological processes are summarized in Table 1, Table 2, Table 3 and Table 4. Since cytokines are the main mediators of inflammation, we furthermore wanted to investigate potentially modified cytokine signatures (Figure 4E). This showed that the majority of cytokines were downregulated by siponimod in LPS-treated microglia with the most notable genes being Ifnb1, Ita, Tnfsf14, Il16, Cxcl13 and Cxcl10. In contrast, we found that Faslg, Il1rn and Tnfsf18 were significantly upregulated.

## 3. Discussion

There is conclusive evidence that siponimod exerts beneficial effects on different aspects of EAE, an established animal model for MS. It decreases disease severity, the degree of demyelination and improves cortical network functionality [4,5]. However, the exact cellular mechanisms underlying these effects remain largely elusive. The same is true for siponimod’s effects on disease progression and brain atrophy in secondary progressing (SP)MS [3]. In this regard, it is important to mention that fingolimod, another S1PR modulator, has already been shown to modulate microglial activation [25,26] which could be further corroborated by PET-CT imaging of MS lesions [27]. Given its more specific receptor profile in comparison to fingolimod, it is of great interest to investigate whether siponimod exerts similar functions.

Our findings now show that in an inflammatory milieu, siponimod (i) protects microglia from adopting activated cytoskeletal architecture and morphologies, (ii) reduces iNOS protein expression, (iii) modulates microglial cytokine expression and (iv) downregulates microglial immunological pathways. These results point to specific effects of siponimod on microglial cells. So far, studies investigating the role of microglia in this context solely used immortalized cell lines. In contrast, this is now the first study to provide data generated in primary rat microglial cells.

Our results indicate that siponimod prevents TNFα upregulation in an inflammatory milieu. TNFα is a pleiotropic pro-inflammatory cytokine which is, amongst other glial cells, produced by microglia [28]. It potentiates glutamate excitotoxicity [29], increases neuroinflammatory responses [30], impairs oligodendroglial differentiation [31] and even induces oligodendrocyte cell death [32]. In MS, TNFα is present in active lesions and its level in the cerebrospinal fluid (CSF) is correlated with disease severity and progression [19]. However, despite beneficial effects of a neutralization of TNFα in animal models its inhibition in the clinical context has resulted in increased disease activity and lesion load progression [33]. This underlines the limited transferability of results generated in in vitro/animal models to the human paradigm. Another cytokine the upregulation of which was reduced by siponimod under inflammatory conditions was IL-1β. This cytokine is present in MS lesions [34,35] and the CSF of MS patients [36,37,38] where its levels are correlated with the number and volume of MS lesions [38]. In EAE, IL-1β is mostly produced by myeloid cells infiltrating the CNS [34] and contributes to leukocyte recruitment and BBB disruption [35].

Interestingly, we also found that siponimod prevented an LPS-induced induction of interferon-β. This is of particular interest as this molecule was the first to be used as an MS drug even though its exact mode of action is not yet fully understood. In general, the beneficial clinical treatment effect is considered to be related to several overlapping mechanisms in the peripheral immune system. This includes the down-regulation of the major histocompatibility complex (MHC) class II expression present on the antigen-presenting cells (i.e., dendritic cells, Langerhans cells and B-cell lymphocytes), the induction of T-cell production of interleukin 10 (IL-10), which shifts the balance toward the anti-inflammatory T helper (Th)-2 cells and the inhibition of T-cell migration [39]. At the same time, only a few studies have investigated the direct impact of interferon-β on brain-resident cells [40,41]. However, these studies were performed exclusively in transgenic animal models and partly contradict each other so that the role of interferon-β is still not entirely clarified.

Finally, siponimod also prevented LPS-induced upregulation of IL-10. IL-10 is a potent anti-inflammatory cytokine which is, among other cell types, expressed by microglia [42,43]. It decreases the release of TNFα, IL-1β, IL-6, IL-8, IL-12 and IL-23, ameliorates the course of EAE, reduces the proliferation of TH1 and TH2 cells, decreases antigen presentation of monocytes and macrophages and has the capacity to act in a neuroprotective manner [17,22]. In MS, IL-10 secretion is decreased prior to relapse and increased during remission. However, no clinical studies investigating the potential benefit of IL-10 for MS have been conducted so far.

The analysis of the transcriptome of LPS/siponimod co-stimulated microglia vs. LPS only stimulated microglia revealed follistatin (Fst), cyclin dependent kinase inhibitor 1c (Cdkn1c), Cd9 and Cd34, as well as Cxcl11, Cd38 and fascin-1 (Fscn1) among the most significantly differentially expressed genes, all of which are strongly associated with microglial polarization states in different injuries and diseases [44,45,46,47,48,49,50]. This further underlines that siponimod stimulation leads to a specific change in the pattern of microglial gene regulation which cannot be assigned to a “classic” M1 or M2 polarization and appears to be more complex. However, the fact that the top 4 cluster of the gene enrichment analysis related to the 1767 downregulated genes were all associated with immunological functions indicates that it has significant immunomodulatory properties. However, besides the association of downregulated DEGs with immunological functions, we found that siponimod modulates additional intriguing clusters of biological processes. For instance, the upregulated DEGs display enrichments in the response to unfolded protein as well as endoplasmic reticulum (ER) stress which is in line with previous findings linking sphingosine-1-phosphate metabolism to these processes [51]. In this regard, mainly ceramide, one of sphingosine-1-phosphate’s precursors, was found to specifically induce ER stress via the CD95-PERK signalling pathway leading to an increase in unfolded protein [52]. Taken together, this finding suggests that although S1PR signalling is inhibited by siponimod treatment, sphingosine-1-phosphate metabolism remains intact. Furthermore, autophagy, another pathway enriched upon siponimod treatment, is thought to be one of the key regulators of innate immune responses [53]. Apart from that, we found downregulated DEG clusters in endocytosis, one of the key mechanisms of S1PR signalling and clusters in actin cytoskeletal organisation, which we gauge to be in line with our finding that siponimod modulates microglial morphology as shown in Figure 1D.

As stated above, the interplay of different cytokines is an important factor in MS. Therefore, our observation that siponimod creates a completely new cytokine signature in pro-inflammatory microglia is of great interest even though it cannot be characterized within the framework of M1/M2 polarization. In this regard, recent RNA sequencing (RNASeq) and genome-wide association studies (GWAS) suggest the existence of several subgroups of disease-associated microglia (DAMs) [54,55,56,57]. All these DAMs, which were first described in neurodegenerative diseases such as Alzheimer’s disease, show unique transcriptional and functional signatures.

Regarding the limitations of our study, we are aware that the use of bacterial endotoxin as a pro-inflammatory stimulus is debatable. Even though LPS is the most commonly used molecule in this context [58,59] recent studies suggest that in CNS damage cytokines such as interferon γ (IFNγ) and TNFα may be also appropriate [36,60]. For instance, LPS was found to evoke higher pro-inflammatory gene expression but also increased several anti-inflammatory genes which is in line with our finding of an increased IL-10 and IFNβ expression. However, identical to our study, these results were generated in rat microglia so that the translation to the human paradigm is still pending. Furthermore, in this study we only examined parallel stimulation with LPS and siponimod. We selected this approach to generate first insights into the effects of siponimod on a single CNS cell type in a pro-inflammatory milieu to which the RNASeq data additionally contributed. In future studies, it will therefore be of interest to investigate to what degree siponimod (pre)treatment can be protective or whether a delayed application exerts similar rescue effects.

In general, even though we show a direct effect of siponimod on CNS-resident cells, this medication certainly exerts its most profound effects on pro-inflammatory peripheral immune cells. It effectively prevents them from leaving lymphoid tissues and thereby averts CNS immune cell infiltration. However, it is known from post mortem histopathology that microglia, that were primed by invading peripheral immune cells during initial disease relapses, create a milieu of smouldering inflammation behind a closed blood–brain barrier (BBB). As a result, siponimod might exert a beneficial effect on these microglia-mediated processes which are probably not even visible on conventional MRI, let alone clinically. This might also explain why siponimod is effective in active secondary progressive MS. In contrast and correspondingly, it has no effect in primary progressive MS where neurodegeneration is predominant. In conclusion, it is conceivable that via a modulation of microglial behaviour siponimod modulates neuroinflammatory processes in MS. Future studies will have to further define the exact microglial subtype associated with siponimod stimulation and its impact in MS.

## 4. Materials and Methods

### 4.1. Primary Rat Microglial Cell Culture

All animal use complies with the ARRIVE guidelines and were carried out in accordance with the National Institutes of Health guide for the care and use of Laboratory animals (NIH Publications No. 8023, revised 1978). The Institutional Review Board (IRB) of the ZETT (Zentrale Einrichtung für Tierforschung und wissenschaftliche Tierschutzaufgaben) at the Heinrich Heine University Düsseldorf has approved all animal procedures under licences O69/11. Briefly, dissociated P1 Wister rat cortices were cultured on T-75 cell culture flasks in Dulbecco′s Modified Eagle′s Medium (DMEM; Thermo Fisher Scientific, Waltham, UK) substituted with 10% foetal calf serum (FCS; Capricorn Scientific, Palo Alto, CA, USA) and 4 mM L-glutamine (Invitrogen, Carlsbad, CA, USA), 50 U/mL penicillin/streptomycin (Invitrogen, Carlsbad, CA, USA) as previously described [61]. After 10 days, flasks were shaken at 180 rpm/min at 37 °C for 2 h and microglia-containing supernatants were collected. Afterwards, cell suspensions were centrifuged for 5 min at 300× *g* at 4 °C, supernatants were discarded and the pellet was resuspended in 10 mL and plated onto bacterial dishes and kept in the incubator (37 °C, 5% CO_2_ and 90% humidity) allowing for cell attachment to the surface. Culture flasks were again loaded with fresh DMEM medium and shaken for another 22 h at 37 °C in order to increase the final cell yield. Afterwards, supernatants were again transferred to bacterial dishes to allow for attachment. Microglia-containing bacterial dishes from the first and second shaking steps were checked for viability via bright-field microscopy, medium was discarded and cells rinsed with Dulbecco’s Phosphate Buffered Saline (D-PBS). Microglia were dislodged by accutase (Thermo Fisher Scientific, Darmstadt, Germany), which was stopped by FCS-containing DMEM medium. Microglial cell suspensions were then centrifuged for 5 min at 300× *g* at 4 °C and cell-free supernatants were discarded. Afterwards, cell pellets were resuspended in 80 µL MACS buffer containing 0.5% BSA in D-PBS and 20 µL of anti-rat CD11b/c microbeads (Miltenyi Biotec, Bergisch-Gladbach, Germany) were added for 15 min at 2–8 °C to allow for binding. Cells were then washed adding 2 mL of MACS buffer and spun down for 5 min at 300× *g* at 4 °C. Supernatants were again discarded and pellets were resuspended in 500 µL MACS buffer and subjected to MACS-sorting according to the manufacturer’s protocol (Miltenyi Biotec, Bergisch-Gladbach, Germany). The resulting cell suspension was again spun down for 5 min at 300× *g* at 4 °C, pellets were resuspended in 1 mL DMEM and cell viability and numbers were quantified using trypan blue staining. Average cell purities as assessed by Iba1-positivity were consistently around 98%. Microglia were seeded on 8-well Lab-Tek chamber slides for immunocytochemistry or 24-well plates for the analysis of mRNA transcripts at different concentrations in microglia medium (10% FCS, 2 mM L-glutamine, 50 U/mL penicillin/streptomycin in DMEM). For stimulation experiments, siponimod (BAF312; kindly provided by Novartis) was solved at a concentration of 50 mM in dimethyl sulfoxide (DMSO; Sigma-Aldrich, St. Louis, MO, USA) and lipopolysaccharide (LPS; Sigma-Aldrich, St. Louis, MO, USA) was solved in ddH_2_O at a concentration of 1 mg/mL. Both reagents were aliquoted in appropriate amounts and stored at −80 °C. Each aliquot was thawed once and discarded afterwards. One day after microglial isolation, cells were stimulated with 10 or 50 µM siponimod or similar amounts of DMSO with or without 100 ng/mL LPS in microglia media. After 1 and 3 days, respectively, cell cultures were either fixated for follow up immune-cytochemistry or lysed for RNA preparation, cDNA synthesis and qPCR analysis.

### 4.2. Immunocytochemistry

For immunocytochemistry, microglia were fixed for 10 min with 4% paraformaldehyde (PFA), D-PBS washed, blocked for 45 min using 10% normal donkey serum (NDS; Sigma-Aldrich, St. Louis, MO, USA) and 0.5% Triton X-100 (Sigma-Aldrich, St. Louis, MO, USA) in D-PBS. Afterwards, cells were incubated at 4 °C overnight with primary antibody solution containing, 10% NDS and 0,1% Triton X-100 in D-PBS with rabbit anti-Iba1 (1/500, WAKO Pure Chemical Corporation, Osaka, Japan; RRID: AB_839504) and goat anti-iNOS (1/250; Abcam, Cambridge, UK, RRID: AB_301857). Following D-PBS washes secondary donkey anti-rabbit Alexa Fluor 488 and donkey anti-goat Alexa Fluor 594 (1/500; Thermo Fisher Scientific, Darmstadt, Germany) were added for 2 h at room temperature. Nuclei were stained in parallel with 4′, 6-diamidino-2-phenylindole (DAPI; 20 ng/mL, Roche, Basel, Switzerland). Cells were mounted using Citifluor (Citifluor, London, UK) and images were captured on an Axioplan 2 microscope (Zeiss, Jena, Germany) using the same exposure times and light intensities. Phalloidin (Phalloidin CruzFluor 488 Conjugate, Santa Cruz Biotechnology, Dallas, TX, USA) staining was performed according to the manufacturers protocol. Briefly, 1000× phalloidin was diluted 1:1000 together with DAPI (20 ng/mL) in PBS. Prefixed cells were incubated with this solution for 1 h at room temperature. Afterwards, cells were washed 4 times with PCR and mounted using Citifluor (Citifluor, London, UK). The analysis of immune-positive cells was performed on 7 images per well and 2 wells per treatment and replicate, leading to 14 analysed images per treatment and replicate. On average, each image contained ~30 cells resulting in more than 2000 cells per condition being analysed. The quantification was performed using ImageJ software (National Institute of Health (NIH) Bethesda, MD, USA). For the analysis of the mean area per cell, merged images were uploaded in ImageJ software, scale bars were set according to microscope settings and the channels were split. The number of cells was assessed by creating a binary image with a threshold to the DAPI channel (80, 255), applying water shedding and analysing all particles with a size of 150–3000 pixels and a circularity of 0.4–1.00. The total area of Iba1-stainings was analysed again by applying a threshold (30, 255), creating a binary image and measuring the total area of Iba1-positive staining. Afterwards, a ratio of the total area of Iba1-positive staining and the total number of cells was calculated. iNOS positive microglia were quantified manually, using the ImageJ tool “cell-counter”.

### 4.3. RNA Preparation, cDNA Synthesis and Quantitative Reverse Transcription (RT)-Polymerase Chain Reaction (PCR)

RNA preparation, cDNA synthesis and quantitative RT-PCR were performed as recently described [61]. Briefly, total RNA purification from cells was performed using the RNeasy procedure (Qiagen, Hilden, Germany). Isolated RNA was afterwards reverse transcribed using the high-capacity cDNA Reverse Transcription Kit (Thermo Fisher Scientific, Darmstadt, Germany). Quantitative determination of gene expression levels was performed on a 7900HT sequence detection system (Thermo Fisher Scientific) using Power SybrGreen PCR master mix (Thermo Fisher Scientific) [62,63]. Following primer sequences were generated via PrimerExpress 2.0 software (Applied Biosystems/Thermo Fisher Scientific, Darmstadt, Germany), as well as tested and determined: Gapdh forward: GAA CGG GAA GCT CAC TGG C, Gapdh reverse: GCA TGT CAG ATC CAC AAC GG, Il1b forward: GAA ACA GCA ATG GTC GGG AC, Il1b reverse: AAG ACA CGG GTT CCA TGG TG, Il10 forward: CCC AGA AAT CAA GGA GCA TTT G, Il10 reverse: CAG CTG TAT CCA GAG GGT CTT CA, Ifnb1 forward: TGG AAG GCT CAA CCT CAG CTA, Ifnb1 reverse: GGG TGC ATC ACC TCC ATA GG, Tnf forward: AGC CC TGG TAT GAG CCC ATG TA, Tnf reverse: CCG GAC TCC GTG ATG TCT AAG T. Gapdh, which proved to be the most accurate and stable normalization gene among a number of others, such as Hprt1, Odc and Tbp, was used as reference gene. Relative gene expression levels were determined according to the ΔΔCt method (Thermo Fisher Scientific) and each sample was measured in duplicate.

### 4.4. Bulk RNA Sequencing

To generate RNASeq data, RNASeq libraries were prepared from DNase digested total RNA samples quantified by Qubit RNA HS Assay (Thermo Fisher Scientific) and capillary electrophoresis using the Fragment Analyzer and the “Total RNA Standard Sensitivity Assay” (Agilent Technologies, Inc. Santa Clara, CA, USA). All samples in this study showed high quality RNA Quality Numbers (RQN; mean = 9.8). The library preparation was performed according to the manufacturer’s protocol using the ‘VAHTS™ Universal RNA-Seq Library Prep Kit for Illumina^®^ V6 with mRNA capture module’. Briefly, 150 ng total RNA were used for mRNA capturing, fragmentation, the synthesis of cDNA, adapter ligation and library amplification. Bead purified libraries were normalized and finally sequenced on the HiSeq 3000/4000 system (Illumina Inc. San Diego, CA, USA) with a read setup of SR 1 × 150 bp. The bcl2fastq tool (v2.20.0.422) was used to convert the bcl files to fastq files as well for adapter trimming and demultiplexing. Data analyses on fastq files were conducted with CLC Genomics Workbench (version 22.0.2, QIAGEN, Venlo. NL). The reads of all probes were adapter trimmed (Illumina TruSeq) and quality trimmed (using the default parameters: bases below Q13 were trimmed from the end of the reads, ambiguous nucleotides maximal 2). Mapping was performed against the Rattus norvegicus (mRatBN7.2.106; 5 July 2022) genome sequence. After grouping of samples (three biological replicates each) according to their respective experimental condition, the statistical differential expression was determined using the Differential Expression for RNA-Seq tool (version 2.6, CLC Genomics Workbench). The resulting *p* values were corrected for multiple testing by FDR and differentially expressed genes (DEGs) were filtered setting a threshold at the FDR adjusted *p*-value of 0.05 and a fold-change of ±1.5. The Gene Set Enrichment and Pathway analysis of differentially up- and downregulated genes was performed using Metascape platform using default parameters (*R. norvegicus*; 1 August 2022).

### 4.5. ELISA

To assess the microglial secretion of TNFα, IL-1β, IL-10 and IFNβ, the culture medium was harvested, spun down at 1000× *g* for 5 min at 4 °C, frozen on dry ice and stored at −80 °C. On the day of analysis, all reagents were thawed and adjusted to RT before the culture medium was measured in duplets using the following colorimetric sandwich ELISA kits according the manufacturers protocol: rat TNF alpha ELISA Kit (ab100785, Abcam), rat IL-1 beta ELISA kit (ab100768, abcam), rat IL-10 ELISA Kit (ab100764, abcam), Rat IFN-beta ELISA Kit (NBP3-06753, Novus Biologicals). After the generation of a 4-parameter logistic standard curve using Graph Pad Prism 8.4.3 (GraphPad Software, San Diego, CA, USA), total protein concentrations were calculated. Protein levels below the detection limits of the used ELISAs were set to 0.

### 4.6. Statistical Analysis

Data are presented as mean values ± standard error of the mean (SEM). All data passed the Shapiro–Wilk test for normality. Therefore, significance was assessed either by 1-way analysis of variance (ANOVA) followed by Tukey’s post hoc test or by an unpaired students *t*-test both using Graph-Pad Prism 8.4.3 (GraphPad Software, San Diego, CA, USA). The experimental groups were considered significantly different at * *p* < 0.05, ** *p* < 0.01, *** *p* < 0.001. n represents the number of independent experiments.

## Figures and Tables

**Figure 1 ijms-23-13278-f001:**
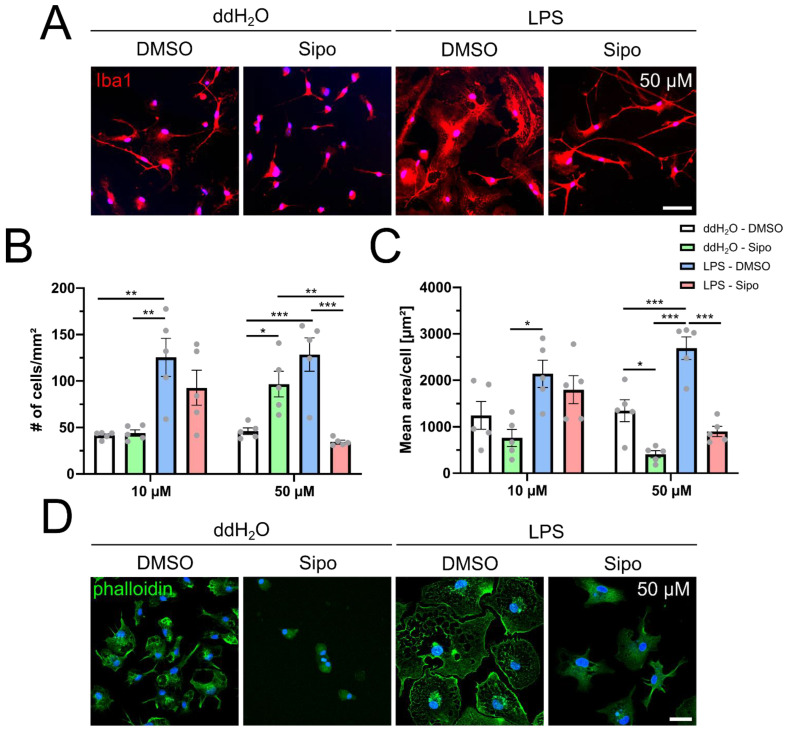
Siponimod modulates morphology and actin cytoskeleton organization of primary rat microglia in an inflammatory milieu. (**A**) Representative images of +/− 50 µM siponimod +/− LPS stimulated microglia after 3 days stained against Iba1 and DAPI. (**B**) Quantification of microglial cell numbers +/− 10/50 µM siponimod +/− LPS after 3 days. (**C**) Quantification of the mean cell area of microglial cells by dividing the total area of Iba1 staining by the cell counts from microglia stimulated for 3 days with +/− 10/50 µM siponimod +/− LPS. (**D**) Evaluation of cytoskeletal organization via Alexa488-coupled phalloidin of 3 days stimulated microglia (+/− 50 µM siponimod, +/− LPS). Data are presented as mean values ± SEM. Grey dots represent individual data points. Significance was assessed by 1-way analysis of variance (ANOVA) followed by Tukey’s post hoc test using Graph-Pad Prism 8.4.3 (GraphPad Software, San Diego, CA, USA). The experimental groups were considered significantly different at * *p* < 0.05, ** *p* < 0.01, *** *p* < 0.001. Scale bar: (**A**) 50 µm, (**D**) 25 µm.

**Figure 2 ijms-23-13278-f002:**
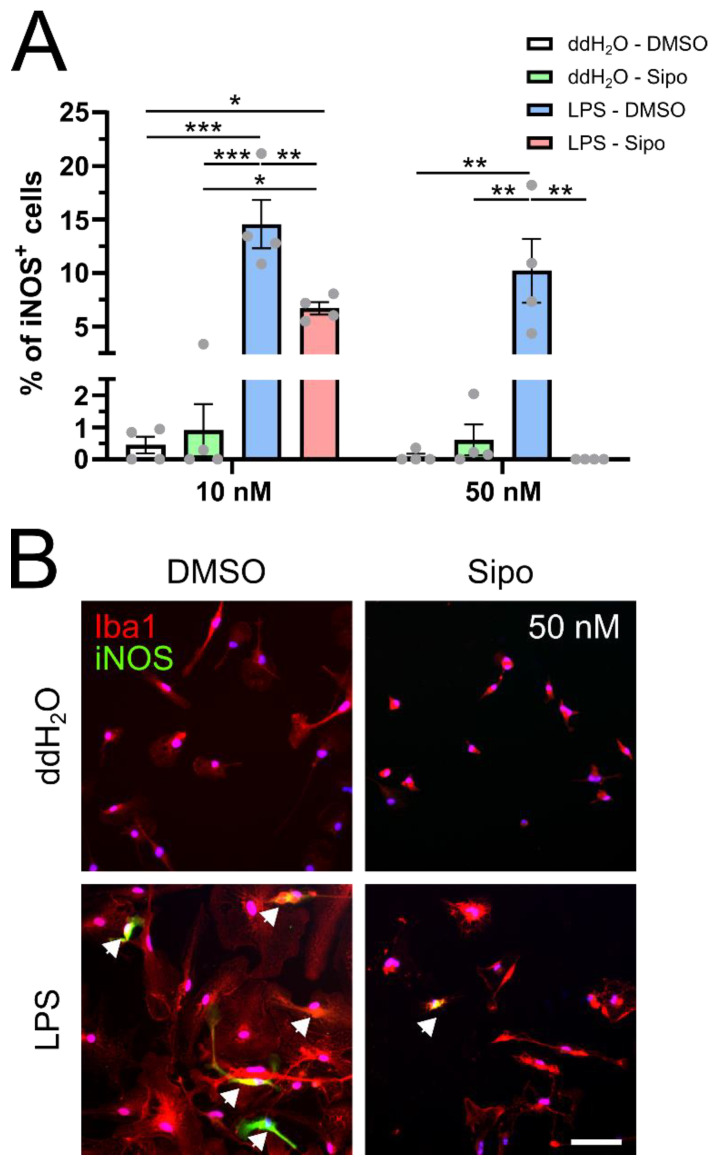
Siponimod reduces microglial iNOS protein expression in an inflammatory milieu. (**A**) Quantification of the percentage of iNOS positive microglia after 3 days of stimulation with +/− 10/50 µM siponimod +/− LPS. (**B**) Representative images of Iba1, iNOS co staining after stimulation for 3 days with +/− 50 µM siponimod +/− LPS. Data are presented as mean values ± SEM. Grey dots represent individual data points. Significance was assessed by 1-way analysis of variance (ANOVA) followed by Tukey’s post hoc test using Graph-Pad Prism 8.4.3 (GraphPad Software, San Diego, CA, USA). The experimental groups were considered significantly different at * *p* < 0.05, ** *p* < 0.01, *** *p* < 0.001. Scale bar: 50 µm.

**Figure 3 ijms-23-13278-f003:**
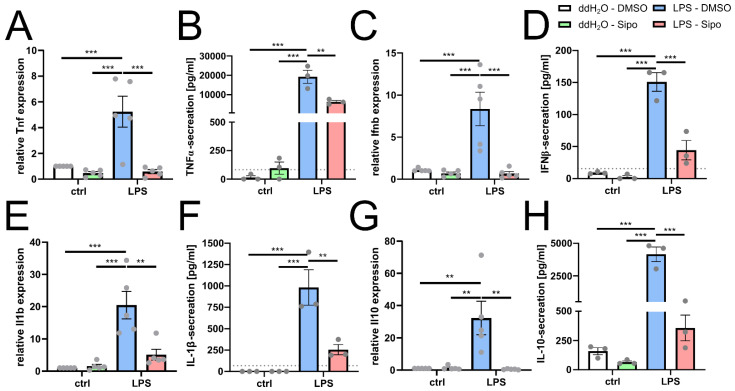
Siponimod modulates microglial cytokine expression and secretion in an inflammatory milieu. (**A**,**C**,**E**,**G**) quantitative RT-PCR analysis of Tnf (**A**), Ifnb (**C**), Il1b (**E**) and Il10 (**G**) gene expression after stimulation for 1 day with +/− 50 µM siponimod +/− LPS. (**B**,**D**,**F**,**H**) Quantification of TNF-α (**B**), IFNβ (**D**), IL-1β (**F**) and IL-10 (**H**) protein concentration in the cell culture medium of microglial cells stimulated for 3 days with +/− 50 µM siponimod +/− LPS using respective quantitative sandwich ELISA assays. Grey-dotted lines indicate the lowest standard of the ELISA kit: TNFα = 82.3 pg/mL, IFNβ = 15.63 pg/mL, IL-1β = 68.59 pg/mL and IL-10 = 8.23 pg/mL. Data are presented as mean values ± SEM. Grey dots represent individual data points. Significance of gene expression analysis as well as ELISA was assessed by 1-way analysis of variance (ANOVA) followed by Tukey’s post hoc test using Graph-Pad Prism 8.4.3 (GraphPad Software, San Diego, CA, USA). The experimental groups were considered significantly different at ** *p* < 0.01, *** *p* < 0.001.

**Figure 4 ijms-23-13278-f004:**
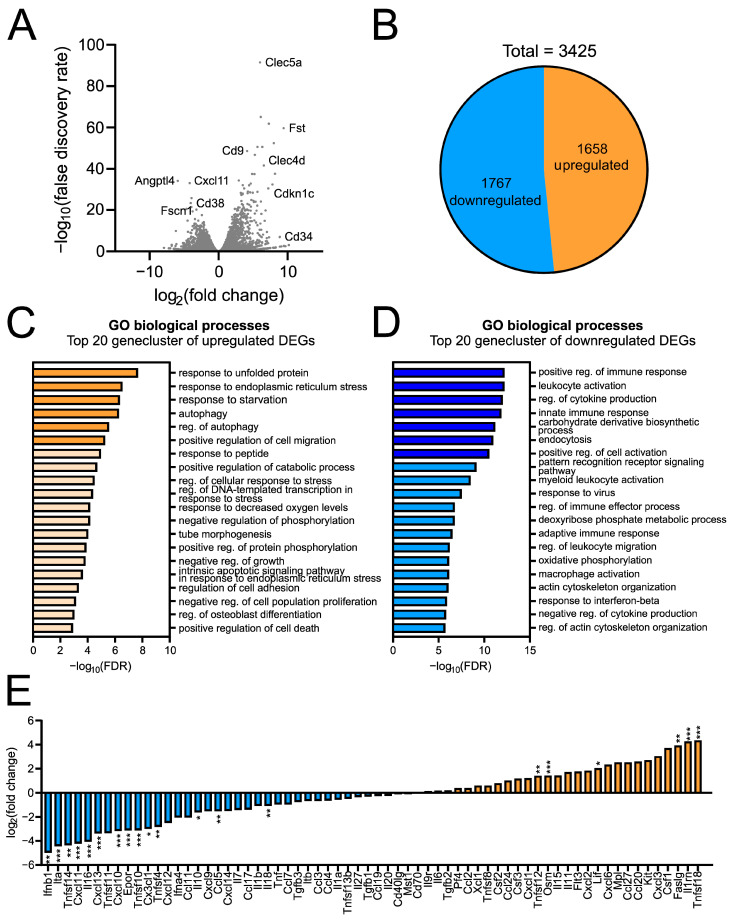
Siponimod alters the immunological signature of LPS-stimulated microglia. Volcano plots showing log_2_(fold-change) against log_10_(false discovery rate) for the comparisons of LPS/siponimod costimulated versus LPS-stimulated microglia (**A**). Identification of 3425 DEGs (fold change of ±1.5 and an FDR adjusted *p*-value of ≤0.05), of which 1658 were up- and 1767 were downregulated (**B**). Gene Set Enrichment and Pathway analysis of up- (**C**) and downregulated (**D**) DEGs in order of their −log_10_(FDR) according to the Benjamin and Hochberg adjustment. Cytokine signature of LPS-siponimod co-stimulated compared to LPS alone (**E**). Significance of transcriptome analysis was assessed by the RNA-Seq tool (version 2.6). The experimental groups were considered significantly different at * *p* < 0.05, ** *p* < 0.01, *** *p* < 0.001.

**Table 1 ijms-23-13278-t001:** Positive regulation of immune response.

Gene Symbol	log_2_ Fold Change	*p* Adjusted
Lta	−4.42	2.07 × 10^−6^
Cd38	−4.04	4.58 × 10^−24^
C3ar1	−2.69	8.27 × 10^−10^
Pycard	−2.69	8.97 × 10^−8^
Card11	−2.59	1.60 × 10^−12^
Tifa	−2.53	3.81 × 10^−13^
Cd180	−2.47	1.37 × 10^−6^
Nlrp3	−2.10	2.02 × 10^−13^
Tfrc	−1.91	1.31 × 10^−6^
Lat2	−1.82	3.68 × 10^−11^
Slamf1	−1.76	5.45 × 10^−6^
RT1-N3	−1.76	8.05 × 10^−8^
Cmklr1	−1.74	2.19 × 10^−5^
Ctsh	−1.72	4.42 × 10^−6^
Cd81	−1.71	1.79 × 10^−5^
Dhx58	−1.65	4.29 × 10^−6^
Ada	−1.60	3.02 × 10^−6^
Xrcc5	−1.59	1.21 × 10^−6^
Lacc1	−1.57	3.02 × 10^−6^
Trex1	−1.57	1.51 × 10^−6^
Nectin2	−1.52	2.44 × 10^−5^
Cyrib	−1.36	5.69 × 10^−6^
Tlr6	−1.32	3.37 × 10^−6^
Nod1	−1.29	8.86 × 10^−6^
Ifi35	−1.25	7.32 × 10^−6^

**Table 2 ijms-23-13278-t002:** Leukocyte activation.

Gene Symbol	log_2_ Fold Change	*p* Adjusted
Il21r	−3.66	1.46 × 10^−7^
P2ry12	−3.44	4.64 × 10^−7^
Cxcl13	−3.35	7.65 × 10^−7^
Fgl2	−3.20	4.82 × 10^−11^
Cxcl10	−3.14	1.62 × 10^−10^
Cd244	−2.94	1.36 × 10^−15^
Slamf9	−2.88	2.15 × 10^−7^
Tnfrsf11a	−2.81	1.94 × 10^−7^
Klf2	−2.81	2.27 × 10^−7^
Pycard	−2.69	8.97 × 10^−8^
Anxa3	−2.60	2.16 × 10^−7^
Card11	−2.59	1.60 × 10^−12^
Clec4a3	−2.54	7.55 × 10^−7^
Cd180	−2.47	1.37 × 10^−6^
Trpm2	−2.43	6.88 × 10^−13^
Sh3pxd2a	−2.37	1.25 × 10^−7^
Actb	−2.33	5.90 × 10^−8^
Il6r	−2.30	5.13 × 10^−10^
Lrrk1	−2.09	1.13 × 10^−6^
Tfrc	−1.91	1.31 × 10^−6^
Axl	−1.86	5.20 × 10^−8^
Lat2	−1.82	3.68 × 10^−11^
Ubash3b	−1.61	4.55 × 10^−8^
Xrcc5	−1.59	1.21 × 10^−6^
Prdx2	−1.35	1.84 × 10^−7^

**Table 3 ijms-23-13278-t003:** Regulation of cytokine production.

Gene Symbol	log_2_ Fold Change	*p* Adjusted
Cxcl11	−4.20	9.69 × 10^−34^
Il16	−4.05	5.43 × 10^−7^
P2ry12	−3.44	4.64 × 10^−7^
Cxcl13	−3.35	7.65 × 10^−7^
Siglec8	−3.28	1.76 × 10^−9^
Fgl2	−3.20	4.82 × 10^−11^
Cxcl10	−3.14	1.62 × 10^−10^
Cd244	−2.94	1.36 × 10^−15^
Tnfrsf11a	−2.81	1.94 × 10^−7^
Klf2	−2.81	2.27 × 10^−7^
Samhd1	−2.72	2.29 × 10^−7^
C3ar1	−2.69	8.27 × 10^−10^
Pycard	−2.69	8.97 × 10^−8^
Card11	−2.59	1.60 × 10^−12^
Clec4a3	−2.54	7.55 × 10^−7^
Dagla	−2.32	2.04 × 10^−7^
Il6r	−2.30	5.13 × 10^−10^
Cxcl17	−2.28	8.05 × 10^−8^
Nlrp3	−2.10	2.02 × 10^−13^
Ezr	−1.91	1.30 × 10^−7^
Ccr5	−1.90	1.45 × 10^−7^
Axl	−1.86	5.20 × 10^−8^
Ctnnbip1	−1.77	1.13 × 10^−6^
Nt5e	−1.48	2.00 × 10^−7^
Prdx2	−1.35	1.84 × 10^−7^

**Table 4 ijms-23-13278-t004:** Innate immune response.

Gene Symbol	log_2_ Fold Change	*p* Adjusted
Cx3cr1	−3.68	2.18 × 10^−6^
Wfdc21	−3.22	6.08 × 10^−10^
Evl	−3.17	1.72 × 10^−11^
Cxcl10	−3.14	1.62 × 10^−10^
Clec4a1	−2.78	6.22 × 10^−7^
Pycard	−2.69	8.97 × 10^−8^
Ccl12	−2.59	1.23 × 10^−6^
Clec4a3	−2.54	7.55 × 10^−7^
Tifa	−2.53	3.81 × 10^−13^
Mrc1	−2.43	4.99 × 10^−10^
Gbp4	−2.34	4.46 × 10^−9^
Fes	−2.33	1.35 × 10^−6^
Ly86	−2.10	2.17 × 10^−5^
Nlrp3	−2.10	2.02 × 10^−13^
Lrrk1	−2.09	1.13 × 10^−6^
Aif1	−1.90	5.24 × 10^−6^
Coro1a	−1.73	4.18 × 10^−5^
Dhx58	−1.65	4.29 × 10^−6^
Sla	−1.59	5.03 × 10^−6^
Trim25	−1.58	1.22 × 10^−6^
Trex1	−1.57	1.51 × 10^−6^
Tmem106a	−1.43	1.22 × 10^−5^
Tlr6	−1.32	3.37 × 10^−6^
Nod1	−1.29	8.86 × 10^−6^
Ifi35	−1.25	7.32 × 10^−6^

## Data Availability

RNASeq datasets generated for this study are publicly available in the NIH Gene Expression Omnibus (GEO) repository: https://www.ncbi.nlm.nih.gov/geo/, accession number GSE216804.

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
