# Peer review of "Siponimod Modulates the Reaction of Microglial Cells to Pro-Inflammatory Stimulation"

_ijms, 2022, doi:10.3390/ijms232113278_

Round 1
Reviewer 1 Report (Previous Reviewer 2)
In their revised work, the authors show an inhibiting effect of the S1PR modulator siponimod on in vitro microglia cell activation and proinflammatory gene expression. They have now also included data from bulk RNA sequencing showing that inflammatory signature of LPS-treated microglias is reduced by siponimod.
The main problem and concern of this study is the still exorbitantly high concentrations of Siponimod used. The authors have not addressed this point of concern and rather refer to a publication and purporting that such high concentrations as 10uM and 50uM of siponimod were measured in vivo in brain extracts of mice and rats. However, the publication of Bigaud et al (Ref 13) is not supporting this. Actually, in that study, at the high dose of 10mg/kg/d siponimod, maximally 3uM was detected in brain. Using 1mg/kg/d in an EAE model, less than 1uM was detected in brain.
In the setting here in microglial cells in culture, the S1P1 receptor is freely available for the agonist given to the medium and no issue of tissue penetration problem can justify the high concentration used. Therefore, it is mandatory to show a dose-response experiment with siponimod, including the low nM range, for example 1nM (which is 3 times the EC50) up to 50uM. If concentrations of 10xEC50 are inactive, there is a very high probability that the here observed effects are unrelated to S1P1 activation/downregulation and of unspecific nature. Alternatively, use a S1P1-specific antagonist to show that the siponimod-mediated effect on any of the responses (cell morphology or iNOS) is reverted.
Author Response
Please see the attachment

Reviewer 2 Report (New Reviewer)

Round 2
Reviewer 1 Report (Previous Reviewer 2)
The authors have still not shown a pharmacologically convincing effect of siponimod on S1P1 receptor activation in microglia cells by performing a dose- response experiment.
The arguement that the deadline for revision is too short for performing a simple dose-response experiment is not acceptable.
Author Response
Reviewer 1:
The authors have still not shown a pharmacologically convincing effect of siponimod on S1P1 receptor activation in microglia cells by performing a dose- response experiment.
The arguement that the deadline for revision is too short for performing a simple dose-response experiment is not acceptable.
Response: Unfortunately, we still disagree with reviewer 1 on this matter. As pointed out before, Bigaud et al. as well as Gardin et al. support our choice of concentrations as also corroborated via repeated personal communications with Dr. Bigaud. Our concentrations are plausible in the brain, adequate for S1P1 receptor activation and can hence explain our experimental observations. In reiteration of our previous statement, we would like to point out that CSF siponimod levels were found to be 1000 times lower than brain parenchymal siponimod levels. This corresponds to the exact proportion bound to human blood serum (Gardin et al., 2017). Moreover, Fig. 1C and Fig. 2B show dose-dependent effects even with the higher siponimod concentration. Based on these data and the basic fact that it is established that siponimod binds to S1P1 and S1P5, we cannot see, at the current time, why additional receptor blocking experiments would be necessary.
Reviewer 2 Report (New Reviewer)
- Why are control and Siponimod treatments missed in figure 3B, 3D, 3F and 3H?
Response: Within the experimental setup of our ELISA assays (Fig. 3B, 3D, 3F and 3H), we have indeed also measured protein levels of control- and siponimod only-treated cells.However, the detected cytokine levels were below the detection limit of the kit manufacturer so that we decided not to include this data in our manuscript. Moreover, we had to use pure cell culture supernatants in order to properly detect levels of TNF., IFN., IL-1. and IL-10 protein levels which made it impossible to further increase protein concentrations. Of course we agree with the reviewer that this information is important in order to provide proper scientifictransparency and we have therefore changed our manuscript accordingly (line 445 – 448).
Reviewer comment: I still think that it will be better to show the results of control and Siponimd treatments instead of deleting them because these two are essential conditions.
Author Response
Reviewer 2:
Reviewer comment: I still think that it will be better to show the results of control and Siponimd treatments instead of deleting them because these two are essential conditions.
Response: We appreciate this constructive remark and adapted the figure and figure legend (Fig. 3) as well as material and method and results section accordingly (line 143-145, 450).
Round 3
Reviewer 1 Report (Previous Reviewer 2)
There remains an unsolved debate about the interpretation of the data and personal communications are no proof of validity of the data.
This manuscript is a resubmission of an earlier submission. The following is a list of the peer review reports and author responses from that submission.
Round 1
Reviewer 1 Report
While the paper on the SPMS study of siponimod does say there was a beneficial effect in SPMS, clinical and imaging, there is data, used by at least one regulatory agency that the benefits were basically in those patients who had “active” MS. That implies that the effect is still predominately due to the peripheral immune effects not the effects shown mainly in in vitro and EAE experiments, rather than on endogenous CNS cells. Admittedly siponimod does enter the CNS but so does fingolimod, which binds to S1P1 and S1P5, as well as S1P3 and 4, and has in vitro and EAE beneficial effects, had no effect in PPMS, which is part of the MS spectrum, not a separate disease unto itself. Would the authors comment on these issues in Introduction or preferably in Discussion? It helps in keeping things in clinical prospective.
Cytokines that re representative of Th2 cell activity upregulates the gene for arginase and suppress the activation of the gene for iNOS in glial cultures in vitro and IL4 does this in immune cells. The opposite effect is seen with IL12. Did you look at IL12? Why do you think you didn’t see effects on arginase or IL10?
Despite the increase in TNF-a in MS lesions and known in vitro effects of TNF-a, inhibition of TNF-a worsened S and can induce first time CNS and PNS inflammatory demyelinating syndromes. While this might relate to differences in the 2 receptors for TNF-a in some species, it is worth reminding readers of this difference of in vitro/animal models and clinical MS.
Did you examine the effects of LPS, siponimod and siponimod on the LPS effect on polarization of the microglia in your system? Likely of importance in the different disease activity vs. protective and reparative effects of microglia, as you point out in Introduction.
While fingolimod and siponimod have beneficial effects in EAE, can those studies, even if done with passive transfer EAE, definitively separate the peripheral immune effects from direct effects on endogenous CNS cells? And despite the effects of fingolimod on PET assessment of microglial activity, fingolimod had no real beneficial effects in PPMS. This dichotomy of in vitro and EAER effects in PPMS and SPMS in the absence of ’active’ disease needs to be mentioned and discussed for clinical vs. experimental prospective.
Reviewer 2 Report
In this study, the authors report on the in vitro effect of the novel S1PR modulator siponimod (BAF312) on microglia activation and proinflammatory gene expressions. They isolated from rats primary microglia cells that were then stimulated with LPS as a proinflammatory agens to show increased cell area as a measure of activated microglia. In the presence of 50uM of siponimod, the increased cell area was normalized. Furthermore, the proinflammatory genes TNFa, IL-1b, and iNOS were reduced by siponimod. Surprisingly, also the anti-inflammatory genes IL-10 and IL-27 were also downregulated by siponimod.
Major concerns:
1.The concentration of siponimod used in this study (10uM and 50uM) is far too high. Considering that the EC50 of Siponimod for S1PR1 is 0.3nM, these exorbitantly high doses are not appropriate and will probably show additional unspecific effects not attributable to S1P1. In Fig. 1A, a significant reduction of the cell area was only seen at 50uM, i.e. 100’000 fold higher than the EC50. The authors must show a detailed dose response curve for the effects on microglia activation and the various parameters and only if the active concentrations are in a reasonable range (between 1nM and 100nM), the effect likely mediated by S1P1 and of clinical relevance.
2.In Fig. 2, they show the mRNA expression of various genes, while in Fig. 3, they show iNOS protein by immunofluorescence staining. The authors should stay consistent and show the protein expression of all genes which is more conclusive than mRNA expression.
- In the discussion part (lines 152-158), they discuss the effect of nitric oxide on the BBB and propose that the reduced NO production in their experiments may act protective. However, they provide no evidence that NO is indeed reduced in their setting. They should analyze nitrite formation in the supernatants of their cells, again including the whole concentration range.
4.Results on IL-10 mRNA in Fig. 2C shows that this anti-inflammatory cytokine is also downregulated by siponimod. This makes no sense, but may be due to the too high concentration of Siponimod that shows unspecific effects.
5.In line 100, they mention that other genes, like IL-6, IL-4 and arginase, are not regulated. Instead of stating “data not shown”, they should include them in a supplementary file.
Round 2
Reviewer 2 Report
The authors did not take up any of the concerns. The main argument is that the revision timeline does not allow any further experiments.
About the concern of using too high concentrations of siponimod, they argue that at concentrations upto 1uM of siponimod, no effect was observed. Since 1uM is already a maximal concentration for S1P1 activation, it is clear that whatsoever the mechanism of the herein obersved effect on microglia, it is not due to S1P1 activation.